# Trimethoprim: An Old Antibacterial Drug as a Template to Search for New Targets. Synthesis, Biological Activity and Molecular Modeling Study of Novel Trimethoprim Analogs

**DOI:** 10.3390/molecules25010116

**Published:** 2019-12-27

**Authors:** Agnieszka Wróbel, Dawid Maliszewski, Maciej Baradyn, Danuta Drozdowska

**Affiliations:** 1Department of Organic Chemistry, Medical University of Bialystok, 15222 Bialystok, Poland; dawidmaliszewski.dm@gmail.com (D.M.); danuta.drozdowska@umb.edu.pl (D.D.); 2Department of Physical Chemistry, University of Bialystok, Institute of Chemistry, 15245 Bialystok, Poland; m.baradyn@uwb.edu.pl

**Keywords:** trimethoprim analogs, DNA-binding agents, dihydrofolate reductase, molecular docking

## Abstract

A new series of trimethoprim (TMP) analogs containing amide bonds (**1**–**6**) have been synthesized. Molecular docking, as well as dihydrofolate reductase (DHFR) inhibition assay were used to confirm their affinity to bind dihydrofolate reductase enzyme. Data from the ethidium displacement test showed their DNA-binding capacity. Tests confirming the possibility of DNA binding in a minor groove as well as determination of the association constants were performed using calf thymus DNA, T4 coliphage DNA, poly (dA-dT)_2_ and poly (dG-dC)_2_. Additionally, the mechanism of action of the new compounds was studied. In conclusion, some of our new analogs inhibited DHFR activity more strongly than TMP did, which confirms, that the addition of amide bonds into the analogs of TMP increases their affinity towards DHFR.

## 1. Introduction

Folate metabolism has long been recognized as an important and attractive target for the development of therapeutic agents against bacterial, parasitic infections [1], and cancer therapy [2,3]. Dihydrofolate reductase (DHFR) is an essential enzyme, which catalyzes the reduction of dihydrofolate acid (7,8-dihydrofolate, DHF) to tetrahydrofolic acid (5,6,7,8-THF) using reduced nicotinamide adenine dinucleotide phosphate (NADPH) as a cofactor (Scheme 1) [4,5,6]. The crucial role of DHFR is related to biosynthesis pathways of the thymidylate and purines, as well as several other amino acids like glycine, methionine, serine, and N-formyl-methionyl tRNA [7,8,9]. Inhibition of folate-metabolizing enzymes leads to an imbalance in the pathways involved in active synthesizing thymidylate, disrupts DNA replication, and eventually causes cell death [10]. Therefore, DHFR is a very good example of a well-established molecular target of new active compounds, which could be approved as antibacterial drugs and therapeutic agents against a variety of fatal disorders e.g., cancer [11,12]. Several classes of compounds have been explored for their potential antifolate activity; among the most outstanding are diaminopyrimidine [13,14], diaminoquinazolin [15], diaminopteridine [16], and also diaminotriazines [17]. Over the last decade, a lot of research projects have focused on the search for new compounds active against this enzyme, most often derivatives of methotrexate (MTX), which is confirmed to be an effective inhibitor by extensive literature in this field [18,19].

Our last review presents the current state of knowledge on the modification of known DHFR inhibitors as anticancer agents, and shows that multitarget compounds represent a promising approach for discovering new structures for anticancer therapy [20]. Moreover, numerous quotations from literature show the variety of structures bearing 1,3-thiazole [21], 1,3,4-thiadiazole, or 1,2,4-triazole moiety in various fused heterocyclic systems [22,23,24], as well as 1,3,5-triazine [25,26,27,28] or biguanide and dihydrotriazine derivatives [29].

In turn, the most successful inhibitor against bacterial DHFR is trimethoprim (TMP) [2,4-diamino-5-(3,4,5-trimethoxybenzyl) pyrimidine], which is a synthetic, broad-spectrum antimicrobial agent [30]. It is mainly used in the treatment of urinary tract infections, both alone and in combination with a sulfonamide (e.g., sulfamethoxazole, sulfadiazine, sulfamoxole) [31]. This antibiotic is a pyrimidine antifolate drug, which selectively inhibits the bacterial enzyme dihydrofolate reductase (DHFR). The mechanism of this inhibition consists in preventing the conversion of DHF to an active form, i.e., THF [3,32]. One of our reviews presents an extensive range of research literature on the first and most recent achievements in TMP analogs as DHFR inhibitors and underlines new directions in developing and modeling DHFR inhibitors [33]. Currently, Pedrola et al. [34] showed group of TMP analogs display meaningful structural features of the initial drug together with relevant modifications at several points, keeping antibiotic potency and showing satisfactory antimicrobial profile (good activity levels and reduced growth rates), especially against methicillin-resistant *Staphylococcus aureus* (Figure 1). The new products may open new possibilities to fight bacterial infections. The literature analysis confirmed that there are only few reports that would confirm the biological activity of TMP analogs targeting anticancer properties. Singh et al. [35] modified the antibacterial agent TMP to compounds **A** and **B** (Figure 1) with promising anticancer applications. These two compounds had significant tumor growth inhibitory activities over 60 human tumor cell lines and exhibited appreciable interactions with DHFR [34]. Algul et al. [36] have developed a new nonclassical series of propargyl-linked DHFR inhibitors. It was observed that interactions of propargyl-linked inhibitors with Leu22, Thr56, Ser59, Ile60 could potently inhibit human DHFR (*h*DHFR), in contrast to weak inhibition of *h*DHFR by TMP. Based on SARs (structure-activity relationships), Algul et al. [36] reported that hydrophobic substitutions at C6 and the propargylic position increased anticancer potency. Significantly, propargyl-linked compound **C** (Figure 1) exhibited 3500-fold greater potency than TMP [36].

Recent trends in medical chemistry suggest the developing of multitargeting and multifunctional compounds—in addition, it is a worldwide medical research strategy [37]. The term “designed multiple ligands” was coined by Morphy and Rankovic to describe the abovementioned compounds [38]. The mode of designed multiple ligands could offer several potential advantages, such as the increase of therapeutic efficacy, or decrease of cancer drug resistance [39,40].

This work was targeted to design novel candidates for antitumor drugs, which are structurally related to netropsin (NT) and TMP (Figure 2). NT is a natural antibiotic, isolated for the first time by Finlay et al. from the *Streptomyces netropsis* strain [41]. This antibiotic is a classic representative of the group of minor groove binding agents (MGBA) compounds. NT has been classified as an anticancer compound, forming non-intercalating bonds with DNA, but is not used in medicine because of its high cytotoxicity [42]. It was found that the molecules of these antibiotics, which contain amide NH groups, can form hydrogen bonds in the position C-2 of thymine and N-3 of adenine. The direction of amide CONH bonds in both antibiotics is in accordance with the direction of C5′→C3′ of the polynucleotide chain [43]. In addition, the carboxamide group hydrogen binds preferentially at adenine–thymine (A–T) regions, rich in lone pairs of electrons, which act as the hydrogen bond acceptors [44]. However, the antitumor activity of DNA-binding drugs is not only due to their interaction with DNA. Sabry et al. designed and synthesized a new series of hybrid compounds as new structures with more antitumor/or DHFR inhibition activities [45]. Compounds **D** and **F** (Figure 1), containing amide functional groups, showed DHFR inhibitory potency at IC_50_ 0.2 µM, (where MTX IC_50_ = 0.08 µM). Moreover, compound D showed high binding affinity toward amino acid residues Thr56, Ser59, and Ser118 as hydrogen bond acceptors [45]. In turn Wang et al., based on 8,10-dideazaminopterines structures, potent DHFR inhibitors, designed a series of novel compounds (**G**–**I**) (Figure 1) by removing the glutamate moiety and introducing lipophilic groups into the benzene ring [6]. Secondly, these new structures contain an elongated methylene bridge connecting two aromatic rings by one carbon atom, which affects the elasticity of the molecule and adapts to the binding site of enzyme *h*DHFR. Additionally, intramolecular interactions with the amino acids: Asp21, Phe31, Ser59, Ile60 and Pro61 are formed in the active place in the enzyme [6]. These biological results, as well as molecular modeling studies and literature analysis, could be considered as a template for design and synthesis of the new structures. Our novel analogs contain amine bonds in the place of the methylene linker. In addition, we presented the effect of extension of the methylene bridge connecting two aromatic ring by one or two carbon atoms in model-structure TMP. In this paper, we present a synthesis of compounds **1**–**6** (Figure 2), novel TMP analogs, and preliminary research of biological activity. Our investigation includes the DHFR enzyme inhibition test, DNA-binding effects, and molecular docking study. This paper is an original investigation of rational drug design program aiming at the development of TMP analogs as potential antitumor compounds and minor groove binders. We hope that this will broaden the range of biological activity of the new TMP analogs and allow us to obtain new compounds with anticancer activity.

## 2. Results and Discussion

### 2.1. Preparation of TMP Analogs

In this work, we intended to obtain a new series of TMP analogs (**1**–**6**). Solid-phase synthesis seems to be a good method to obtain trimethoprim analogs containing amide bonds. Based on the results presented here, a new series of potentially active substances are planned to be generated using the same method, in order to establish a structure–activity relationship. The presented derivatives **1**–**6** have been received with sufficient efficiency and purity. The chemical structures of novel compounds were proved by NMR and LC–MS analysis. The solid-phase synthesis of the new compounds **1**–**6** shown in Figure 3 was carried out according to the protocol presented earlier for netropsin analogs [46].

### 2.2. Biological Assays

#### 2.2.1. The Ethidium Bromide Assay—DNA-Binding Effects

The ethidium bromide assay showed that the investigated compounds can bind to plasmid DNA (Table 1). The results of this assay are shown as a percentage of the decrease in fluorescence of each substance in relation to control, i.e., netropsin. The DNA-binding effect of NT in the same conditions was 74% [47]. All of these compounds (**1**–**3** and **5**–**6**) respectively (71, 43%; 45, 18%; 69, 92%; 69, 17%; 71, 43%); except **4** (80, 43%) were characterized by a higher binding strength to pBR*322* plasmid, in contrast to TMP and MTX, which do not bind to DNA. Table 1 presents that compounds **2**–**3** and **5** showed a higher binding affinity to pBR*322* plasmid compared to the model compound, NT. In addition, compounds **1** and **6** show a similar degree of DNA binding to NT. These data suggest that the amide bond has been one of the beneficial modifications in our work. In addition, it was observed that compounds **2** and **5**, which contain an elongated methylene bridge connecting two aromatic rings by one carbon atom, presented the highest decrease in fluorescence. We assume that the extension of the methylene bridge by one carbon could affect the elasticity of the molecule and introduce new properties in the TMP-derivative molecules.

#### 2.2.2. Ethidium Displacement Assay—Determination of DNA-Binding Constants

To understand the mechanism by which the prepared compounds could act, the DNA-binding properties of the TMP analogs (**1**–**6**) were examined by the ethidium displacement assay and the determination of values of association constants of drug-DNA complexes using calf thymus DNA, T4 coliphage DNA, poly(dA-dT)_2_, and poly(dG-dC)_2_ [48,49]. The determination of association constants enables the qualification of the potential and selectivity of the interactions between ligands and DNA. The binding affinities of the compounds **1**–**6**, as well as TMP and MTX, were compared to NT (Table 1). The values of association constants demonstrated that each of the tested compounds can bind to all of the studied types of DNA, though with different degrees of strength. The affinity of association constants of compounds **1**–**6** in the range of 2.4–5.9 × 10^5^ M^−1^ indicates moderate interactions with deoxyribonucleic acid from calf thymus. High values of binding constants for T4 coliphage DNA observed for the tested compounds, especially 3 and 5, were similar to those for NT. This confirms their minor-groove selectivity, since it is known that the major groove of T4 coliphage DNA is occupied by α-glycosylation of the 5-(hydroxymethyl)cytidine residues [50], and the high value of the binding constant of ligand for T4 coliphage DNA provides evidence of its minor-groove specificity. These data also indicate, that compounds **1**–**6** have a very weak interaction with GC-pairs. Higher affinity was observed in the case of AT pairs, which is related to the binding of investigated compounds to calf thymus DNA, containing random AT sequences.

#### 2.2.3. Dihydrofolate Reductase (DHFR) Inhibition

As an attempt to reveal the antitumor potency mode of action, DHFR inhibition activity was evaluated. All of the synthesized compounds **1**–**6** were subjected to the DHFR inhibition evaluation assay reported procedure using recombinant human DHFR enzyme [51]. Results were reported as IC_50_ value (Table 1). MTX and TMP were used as reference drugs (IC_50_ = 0.08 µM and 55.26 µM). Analogs **2** and **3** proved to be the most active DHFR inhibitors in this report with IC_50_ values of 0.99 µM and 0.72 µM, respectively. The compound **4** presented moderate activity with a value of IC_50_ equal to 1.02 µM. Compounds **1**, **5**, and **6** were showed to be slightly active, with IC_50_ values of 21.78, 15.94, and 15.09 µM respectively.

#### 2.2.4. Molecular Docking

The calculations have been carried out in the DHFR active site for **1**–**6** analogs of TMP, as well as for the TMP, and MTX molecules for the sake of comparison. The results for the best binding modes are given in Table 1. The molecular docking results show that MTX has the lowest binding energy of all tested molecules, which corresponds to its high inhibition activity. The unmodified TMP molecule has the highest binding energy, which means that the addition of amide bond into the analogs of TMP increases their affinity towards DHFR. The main moieties involved in the interaction between TMP ligands and the receptor are methoxy groups as hydrogen bond acceptors, the amine and peptide groups as hydrogen bond donors, as well as the aromatic rings involved in the π-π interactions. The results for the redocking of MTX and the molecular docking of TMP into human DHFR are provided in Figure 4. MTX, one of the most active DHFR inhibitors, has binding energy of −9.5 kcal/mol and forms seven hydrogen bonds with residues: Ile-7, Glu-30, Gln-35, Asn-64, Arg-70 and Val-115. We can also observe the π-π interaction with Phe-34. This is also true for the unmodified TMP, which in addition to the above, forms six hydrogen bonds with Ile-7, Ser-59, Val-115, Tyr-121 and Thr-146. Despite that, this ligand shows the least affinity towards DHFR with the score of −7.5 kcal/mol.

The results obtained from molecular docking for the TMP analogs show that they have similar binding affinities, ranging from −7.7 to −8.3 kcal/mol. Compound 2 with the score of −8.3 has the lowest binding energy and it forms five hydrogen bonds with following residues: Ala-9 (N-H∙∙∙O, 2.8 Å), Thr-56 (N-H∙∙∙O, 2.6 Å), Tyr-121 (O-H∙∙∙O, 1.9 Å), Asp-145 (N-H∙∙∙O, 2.1 Å) and Thr-146 (O-H∙∙∙N, 2.6 Å). Compound 3 is the second in order of affinity among TMP analogs and it forms two hydrogen bonds: Lys-55 (N-H∙∙∙O, 2.6 Å) and Tyr-121 (O-H∙∙∙O, 2.8 Å). We have also found the π-π interaction between the trimethoxybenzene ring and Phe-34, as well as hydrogen-π interaction between the pyrimidine ring and Ser-59. Analog 4, with the binding energy of −8.0 kcal/mol, forms two hydrogen bonds: Glu-30 (N-H∙∙∙O, 2.6 Å) and Ser-59 (O-H∙∙∙O, 1.8 Å). Structure **5** forms three hydrogen bonds: Glu-30 (N H∙∙∙O, 2.5 Å), Glu-30 (N H∙∙∙O, 2.6 Å) and Thr-146 (O H∙∙∙O, 2.5 Å). Compound **6** forms two hydrogen bonds: Ser-59 (O-H∙∙∙O, 2.3 Å) and Glu-30 (N-H∙∙∙O, 2.4 Å). It is worth noting that in case of the three latter TMP modifications **4**–**6**, we can also observe the π-π interactions between Phe-34 and the benzene ring in ligand. Lastly, compound **1** exhibits two hydrogen bonds with residues: Ala-9 (N-H∙∙∙O, 2.3 Å) and Ser-59 (O-H∙∙∙O, 2.1 Å). All the binding modes for compounds **1**–**6** are pictured in Figure 5.

During our molecular docking studies, it was found that the addition of methylene groups in the chain linking two rings causes an inversion in the binding mode between molecules without methylene links (**1**, Figure 5a) and those with aliphatic chains (**2** and **3**, Figure 5b,c). The best binding mode of molecule **1** is caused by the interaction with residues Ala-9 and Ser-59. The Ala-9 residue forms a hydrogen bond with the peptide group of **1**. In our results we have observed that, due to the elongation of chain in compound **2**, this interaction is not present (Appendix A, for details, see the Appendix A). This causes that structure to be in the third place among all binding modes of **2**, with score of −7.4 kcal/mol (Appendix A). The reversed structure, on the other hand, allows for analog **2** to form more hydrogen bonds, including the interaction between the amide group and Thr-56 (Figure 5b).

It is believed that the most important residues involved in the DHFR inhibition activity are Ile-7, Glu-30, Phe-34, and Val-115 [52,53]. This is reflected in the high binding energy of MTX, which exhibits interactions with these residues. This may also be the cause of the higher inhibition activity of **4** and **6** that was observed in the fluorescence spectroscopy experiment in this study, since they are able to form hydrogen bonds with Glu-30, and all three of them are interacting via π-stacking with Phe-34. Furthermore, the discrepancy in terms of binding affinity between our TMP derivatives is marginal (−8.3 kcal/mol for **2** compared to −7.7 kcal/mol for **1**), making it hard to determine explicitly which one of them is the most active inhibitor, using approximate methods.

## 3. Material and Methods

### 3.1. General Information

All reagents were purchased from Fluka (Sigma-Aldrich sp. z o.o., Poznań, Poland), Merck (Darmstadt, Germany), or Alfa Aesar (Karlsruhe, Germany) and used without further purification. Dichloromethane (DCM) and dimethylformamide (DMF) were stored under 4 Å molecular sieves. 1H-NMR and 13C-NMR spectra were recorded on a Bruker AC 400F spectrometer (Bruker corp., Fällanden, Switzerland) using TMS as internal standard; chemical shifts δ are reported in ppm. Ethidium bromide was purchased from Carl Roth GmbH (Karlsruhe, Germany). Plasmid pBR*322* was purchased from Fermentas Life Science (Vilnius, Lithuania).

### 3.2. General Procedure

4-Nitrophenyl Wang resin **I** (0.5 g; 0.41 mmol; 0.81 mmol/g) in dry DCM (10 mL), and 2-amino-5-nitropyrimidine (0.23 g; 1.64 mmol) or 4-nitroaniline (0.23 g; 1.64 mmol) dissolved in DCM (10 mL) and pyridine (177.22 μL; 2.2 mmol), arranged in parallel reaction vessels, were the substrates of our reactions. Intermediates **II** were reduced by solution of SnCl2 in DMF (1M, 10mL). The next step of preparation was acylation of amine **III** by using the substance **En** [**E0**-ethyl-3,4,5-trimethoxybenzoate] (0.40 g; 1.64 mmol), [**E1**-3,4,5-trimethoxyphenylacetic acid (0.37 g; 1.64 mmol] and [**E2**-ethyl 3-(3,4,5-trimethoxy-phenyl) propionate (0.44 g, 1.64 mmol)]. The reagents **E0**, **E1** and **E2** were dissolved in a mixture of DCM:DMF:NMP (1:1:1) containing TBTU (0.53 g; 1.64 mmol). The coupling reactions were carried out overnight at room temperature to produce the resin-bound compounds **IV**. Each resin-bound intermediate was washed before proceeding to the next stage. In the last stage of the process the resins were dried and treated with TFA/DCM (50:50) [54]. After evaporation of the solvents we yielded the products **V** as glaze solids. The compounds were characterized by GCMS. Their ^1^H and ^13^C NMR spectra were in agreement with the assigned structures and these data are given (CD3OD), LC–MS and analytical HPLC (Phenomenex C18, Jupiter 90 A, 4 micron, 250 × 10 mm; Phenomenex C18, Jupiter 300 A, 5 micron, 250 × 4 mm; solvents: A, 0.1% aqueous TFA; B, 0.1% TFA in acetonitrile, gradient 0% B to 60% B in A in 30 min, flow rate 1 mL/min, monitored at 220 nm) data are given under the name of each compounds.

*N-(2-Aminopyrimidin-5-yl)-3,4,5-Trimethoxybenzamide* (**1**). ^13^C (MeOD): 164.97 (CONH), 161.80 (1C), 159.27 (1C), 125.14 (2CH), 124.3 (1C), 123.14 (1C), 117.50 (2CH), 116.47 (1C), 114.59 (1C), 36.99 (1C, OCH_3_), 35.32 (1C, OCH_3_), 31.68 (1C, OCH_3_). ^1^H (MeOD): 7.98 (s, 2H, Pirym-H), 7.21 (d, 1H, Ar-H), 6.89 (d, 1H, Ar-H), 2.98 (s, 3H, OCH_3_), 2.85 (s, 3H, OCH_3_), 2.69 (s, 3H, OCH_3_). M = 304.306; [M + H]^+^ 304.3; R_t_ 2.15; 0.052 g (41.70% yield).

*N-(2-Aminopyrimidin-5-yl)-2-(3,4,5-Trimethoxyphenyl)Acetamide* (**2**). ^13^C (MeOD): 164.67 (CONH), 155.70 (1C), 149.03 (1C), 128.90 (1CH), 128.14 (1CH), 126.24 (1C), 123.53 (1C), 121.34 (1C), 118.50 (2CH), 116.34 (2C), 36.93 (1C, OCH_3_), 35.37 (1C, OCH_3_), 31.70 (1C, OCH_3_), 28.36 (1C, CH_2_). ^1^H (MeOD): 8.02 (s, 2H, Pirym-H), 7. 71 (d, 1H, Ar-H), 6.81 (d, 1H, Ar-H), 3.02 (s, 3H, OCH_3_), 2.88 (s, 3H, OCH_3_), 2.71 (s, 3H, OCH_3_), 2.62 (s, 2H, CH_2_). M = 318.333; [M + H]^+^ 318.3; R_t_ 2.25; 0.049 g (37.70% yield).

*N-(2-Aminopyrimidin-5-yl)-3-(3,4,5-Trimethoxyphenyl)Propanamide* (**3**). ^13^C (MeOD): 162.35 (CONH), 159.04 (1C), 157.95 (1C) 124.34 (2CH), 122.28 (1C), 120.02 (1C), 117.13 (1CH), 116.16 (1CH), 114.23 (1C), 111.34 (1C), 35.78 (2C, 2OCH_3_), 34.31 (1C, OCH_3_), 30.77 (2C, CH_2_). ^1^H (MeOD): 7.95 (s, 2H, Pirym-H), 7.17 (d, 1H, Ar-H), 6.85 (d, 1H, Ar-H), 2.88 (s, 6H, 2OCH_3_), 2.72 (s, 3H, OCH_3_), 2.54 (t, 2H, CH_2_), 2.50 (t, 2H, CH_2_). M = 332.357; [M + H]^+^ 332.3; R_t_ 2.02; 0.044 g (32.29% yield).

*N-(4-Aminophenyl)-3,4,5-Trimethoxybenzamide* (**4**). ^13^C (CDCl_3_): 164.87 (CONH), 163.26 (1C), 162.98 (1C), 157.40 (1C), 123.81 (1C), 123.37 (2C), 119.73 (2CH), 117.34 (2CH), 116.81 (2CH), 36.94 (2C, OCH_3_), 32.08 (1C, OCH_3_). ^1^H (CDCl_3_): 7.98 (s, CONH), 7.05 (d, 2H), 6.81 (s, 2H, Ar-H; d, 2H, Ar-H), 2.99 (s, 3H, OCH_3_), 2.86 (s, 3H, OCH_3_), 2.70 (s, 3H, OCH_3_). M = 270.332; [M + H]^+^ 270.3; R_t_ 2.15; 0.062 g (50.41% yield).

*N-(4-Aminophenyl)-2-(3,4,5 Trimethoxyphenyl)Acetamide* (**5**). ^13^C (MeOD): 168.87 (CONH), 153.36 (2C), 144.20 (1C), 138. 11 (1C), 129.87 (1C), 128.53 (1C), 122.37 (2CH), 116.53 (2CH), 106.25 (2CH), 61.16 (1C, OCH_3_), 56.76 (1C, OCH_3_), 49.00 (1C, OCH_3_), 36.95(1C, CH_2_)). ^1^H (MeOD): 7.17 (d, 2H), 6.88 (d, 2H, Ar-H), 6.78 (d, 2H, Ar-H), 2.99 (s, 6H, OCH_3_), 2.80 (s, 3H, OCH_3_), 2.68 (s, 2H, CH_2_); M = 284.359; [M + H]^+^ 284.3; R_t_ 2.25; 0.056 g (41.48% yield).

*N-(4-Aminophenyl)-3-(3,4,5-Trimethoxyphenyl)Propanamide* (**6**). ^13^C (MeOD): 171.97 (CONH), 163.28 (2C), 159.04 (1C), 159.15 (1C), 155.53 (1C), 154.51 (1C), 124.87 (1CH), 123.45 (1CH), 117.52 (1CH), 116.24 (1CH), 107.56 (2CH), 61.11 (1C, OCH_3_), 56.61 (1C, OCH_3_), 44.70 (1C, OCH_3_), 35.78(1C, CH_2_), 30.74 (1C, CH_2_). ^1^H (MeOD): 7.21 (d, 1H, Ar-H), 6.89 (d, 1H, Ar-H), 6.69 (d, 2H, Ar-H), 2.98 (tr, 2H, CH_2_), 2.85 (tr, 2H, CH_2_), 2.69 (s, 9H, OCH_3_). M = 298.386; [M + H]^+^ 298.3; R_t_ 2.25; 0.064 g (47.41% yield).

### 3.3. The Ethidium Bromide Assay—DNA-Binding Effects

The effects of the investigated compound **1**–**6** on plasmid pBR*322* were determined in accordance with the procedure described previously [55]. Each well of a 96-well plate was loaded with Tris buffer containing ethidium bromide (0.1 M Tris, 1 M NaCl, pH 8.0, 0.5 mM EtBr final concentration, 100 μL). Plasmid pBR322 15 μg as water solution (0.05 μg/μL) and NT, TMP, MTX, or compound **1**–**6** (1 μL of a 1 mM solution in water) were added to each well, to obtain a 10 μM final concentration. After the incubation at 25 °C for 30 min. the fluorescence was read on an Infinite M200 fluorescence spectrophotometer (TECAN, Männedorf, Switzerland) (ex. 546 nm, em. 595 nm) in duplicate experiments with two control wells (no drug = 100% fluorescence, no DNA = 0% fluorescence). Fluorescence readings are reported as % fluorescence relative to the controls.

### 3.4. The Ethidium Displacement Bromide Assay—Determination of DNA-Binding Constants

The fluorescence of DNA solutions (calf thymus DNA, T4 coliphage DNA, poly(dA-dT)_2_, and poly(dG-dC)_2_ with the investigated compound (final concentrations 10, 50, 100 µM) was measured by the fluorescence spectrophotometer Infinite M200 TECAN at room temperature according to the procedure described above. Then, the concentration, which reduced fluorescence to 50% was determined for each compound. The data points of fluorescence intensity were fitted to the theoretical curves, with one or two different iterative nonlinear least squares computer routines. The apparent binding constant was calculated from KEtBr [EtBr] = Kapp [drug], where [drug] = the concentration of the tested compound at 50% reduction of fluorescence and KEtBr and [EtBr] are known [49]. TMP and MTX were also investigated. Results are reported as percentage of fluorescence decrease in Table 1. The compounds **1**–**6** and their DNA-bound complexes showed neither optical absorption nor fluorescence at 595 nm and did not interfere with the fluorescence of unbound ethidium bromide.

### 3.5. Dihydrofolate Reductase (DHFR) Inhibition Assay

The procedure for detection of DHFR activity inhibition included preparation of a test mixture in a 1.0 mL quartz cuvette for the spectrophotometric method. Assay buffer—1× for DHFR (pH 7.5) and 1.5 × 10^−3^ units of DHFR were added to the appropriate tube and thoroughly mixed. Then MTX and TMP as positive control, the investigated compounds **1**–**6**, as well as 10 mM NADPH stock solution acid (6 µL for each reaction) were added, to obtain final concentrations of 10, 50 and 100 mM. After mixing accurately, the reaction was initiated by adding 10 mM dihydrofolic acid (5 µL for each reaction) and a kinetics program was immediately started. The changes in absorbance (ΔOD/ min) were measured by the spectrophotometer Specord 200Plus (Analytikjena, Germany) at 340 nm and 22 °C, and the kinetic program (reading every 15 s for 2.5 min) according to the instructions supplied with the set and recommended by the producer [51]. Results are reported as IC_50_ (50% inhibition of enzymatic activity) in Table 1.

### 3.6. Molecular Docking

To gain insight into the binding activity of TMP analogs, a molecular docking study has been carried out, using AutoDock Vina (version 1.1.2) software (Scripps Research Institute, La Jolla, CA, USA) in Department of Physical Chemistry, University of Bialystok [56]. Human DHFR structure with MTX ligand co-crystallized has been downloaded from the Protein Data Bank (PDB: 1U72). To prepare the enzyme for docking studies, the ligand and water molecules have been removed from the PDB structure and the polar hydrogen atoms have been added. The search space has been chosen so that it would include the whole active site pocket and allow for the ligand molecule to be positioned freely. The middle of the grid box has been positioned in the active site of DHFR using coordinates x = 30, y = 15 and z = 0, whereas the grid box was chosen to be cubical with the length of an edge of 20 Å.

First, to validate our method, a redocking procedure has been carried out for the MTX molecule with the prepared PDB structure and then compared to the structure from an X-ray diffraction experiment (resolution 1.9 Å) for DHFR with MTX ligand co-crystallized (PDB:1U72) [57]. The conformation with the highest affinity from our docking studies was found to be very similar to the one from experiment (RMSD = 1.043 Å), especially within the active site pocket (Figure 6).

## 4. Conclusions

A series of novel trimethoprim analogs **1**–**6** containing amide bond was synthesized and investigated. Compounds **2**–**3** and **5** were characterized by a higher binding strength to pBR*322* plasmid. In addition, it was observed that compounds **2** and **5**, containing an elongated methylene bridge connecting two aromatic rings, presented the highest decrease in fluorescence. The determination of values of association constants of drug–DNA complexes assay revealed that compounds **3** and **5** showed the high-value binding constants for T4 coliphage DNA and confirm their minor-groove selectivity.

Results obtained from the molecular docking experiment show that the introduction of amide bond into the TMP analogs increases their affinity to the human DHFR compared to the unmodified TMP (Table 1). Additionally, it was found that even though our molecular docking studies showed lower affinity for the **4**–**6** analogs, they are able to interact with the crucial residues Glu-30 and Phe-34. The effect of increasing the size of the aliphatic chain within TMP analogs is not straightforward and requires further investigation. The binding energies of all analogs were significant and only about 1.2 kcal/mol lesser than the known DHFR inhibitor MTX, making these derivatives promising candidates for antimicrobial agents.

The in vitro experimental findings revealed that all the newly designed and synthesized compounds, especially **2**–**3** and **5**, exhibited higher activity against the DHFR enzyme and higher binding affinity than standard TMP. Moreover, they introduce new aspects of biological activity. These results confirmed our assumption of double activity of the synthesized compounds: DNA-binding effect and DHFR inhibitory activity, which is proven by molecular docking studies. We plan further in vitro investigations of activity on the cancer cell lines to confirm their effectiveness and potential use in therapeutic applications.

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
