# Peer review of "Trimethoprim: An Old Antibacterial Drug as a Template to Search for New Targets. Synthesis, Biological Activity and Molecular Modeling Study of Novel Trimethoprim Analogs"

_molecules, 2019, doi:10.3390/molecules25010116_

Round 1

Reviewer 1 Report

Comments and observations to authors

The authors present in this manuscript the synthesis and biological activity of six amide analogues of trimethoprim, a DHFR enzyme inhibitor. The work has been carried out with good scientific rigor, the compounds prepared were well-characterized and the biological activity and docking studies determined in an appropriate manner. Perhaps a limitation could be the small number of analogues prepared (6).However, they are new compounds and all them are active, a fact that makes the work very interesting.

I only have three minor observations before recommending publication in Molecules

1) I am not an expert in English but I manage to identify some grammatical errors throughout the manuscript. I suggest the authors to review this, maybe with the help of a native English-speaking person.

2) In the experimental part, the authors report the high-resolution mass spectra [M + H] only with experimental values. I recommend that theoretical values be included for comparison to get a better idea on the purity of the compounds.

3) In the docking study, the authors mention that the conformation of both the ligand structure (blue) and the experimental one (green) almost overlap. However, they do not show any numerical value (RMSD) that supports this fact. In Figure 6, it does not seem that they "almost overlap" especially in the part of the aliphatic chain of the molecules.

Once the authors address these observations, I recommend publishing this work in Molecules

Author Response

Thank you for your remarks. 

The revised manuscript contains answers to your comments. We hope that these minor revisions were made to the text will allow the work to be published quickly in Molecules. 

1) I am not an expert in English but I manage to identify some grammatical errors throughout the manuscript. I suggest the authors to review this, maybe with the help of a native English-speaking person.

Answer:

The text has been grammatically corrected. Errors have been corrected and marked green.

2) In the experimental part, the authors report the high-resolution mass spectra [M + H] only with experimental values. I recommend that theoretical values be included for comparison to get a better idea on the purity of the compounds.

Answer:

Theoretical values were included for comparison to get a better idea on the purity of the compounds and were presented in Supplementary Material Sections.

3) In the docking study, the authors mention that the conformation of both the ligand structure (blue) and the experimental one (green) almost overlap. However, they do not show any numerical value (RMSD) that supports this fact. In Figure 6, it does not seem that they "almost overlap" especially in the part of the aliphatic chain of the molecules.

Answer:

The value of RMSD for redocking studies is indeed necessary and should have been added. The value has been calculated using PyMol software with the pair_fit keyword. The result is RMSD=1.043 Å. We have also changed the phrase “almost overlap” to “very similar” since the aliphatic chain conformation shows some discrepancy, as it was pointed out.

Reviewer 2 Report

Good manuscript by Wróbel and coworkers. The premise is interesting, the goal challenging and the results attractive. Overall the paper contains valuable elements and may be publishable afetr addressing some points

1- Add a Supporting Information with details and, importantly, copies of the spectroscopic datra for the NMR spectra (to demonstrate purity)

2- Figure 1 and text: Authors may comment other TMP derivatives recently described with vatiations on the pyrimidine moiety (Font Chem 2019, vol 7, article 475). Also, in this figure, compoud E does not appear, and some overlappings appear in compounds G-I

3- Fiure 3. In section a: add the reagent (is not py).
Also the scheme is valid for compounds 1-3. Although similar, compounds 4-6 require distinct figures. Explain of redraw.

4- Table 1. Headings. Explain superindex b. what is n.b.?

5- Curiosity. Table 1 and related text. Compounds 1-3 show more potency than TMP in human DHFR. Have authors thought to test inhibition on bacterial DHFR. Perhaps add a comment on this.

6- Line 143. Correct the typo.  "of compounds 1-6 in the range of 2.4 - 5.9 x 105 M-1". Should be 10 to the 5, not 105.

7- In the docking section, it is very remarkable (and surprising) that a single methylene leads to an inversion in the binding mode between 1 and 2 (figure 5, a and b). Explain with more detail and add justifications.

8- In the experimental section, please, give the yields with significative figures (42%, line 234, and all the rest)

Author Response

Thank you for your remarks. 

The revised manuscript contains answers to your comments. We hope that these minor revisions were made to the text will allow the work to be published quickly in Molecules. 

1- Add a Supporting Information with details and, importantly, copies of the spectroscopic datra for the NMR spectra (to demonstrate purity)

Answer:

Theoretical values were included for comparison to get a better idea on the purity of the compounds and were presented in Supplementary Material Sections.

2- Figure 1 and text: Authors may comment other TMP derivatives recently described with vatiations on the pyrimidine moiety (Font Chem 2019, vol 7, article 475). Also, in this figure, compoud E does not appear, and some overlappings appear in compounds G-I

Answer:

We comment other TMP derivatives recently described with vatiations on the pyrimidine moiety (Font Chem 2019, vol 7, article 475). We correct Figure 1.

3- Fiure 3. In section a: add the reagent (is not py).
Also the scheme is valid for compounds 1-3. Although similar, compounds 4-6 require distinct figures. Explain of redraw.

In  section a: we add the reagent – pirymidine.

Figure 3 was corrected and presented synthesis for 1-6 compounds.

4- Table 1. Headings. Explain superindex b. what is n.b.?

Answer:

Superindex b and n.b were marked as,,*’’ in Table1 and detailed below the table.

5- Curiosity. Table 1 and related text. Compounds 1-3 show more potency than TMP in human DHFR. Have authors thought to test inhibition on bacterial DHFR. Perhaps add a comment on this.

Answer:

In the future, we plan to conduct research on the antibacterial properties of the obtained compounds (1-6 TMP) e.g. test inhibition on bacterial DHF. However we are focusing on the anticancer  properties as new target.

6- Line 143. Correct the typo.  "of compounds 1-6 in the range of 2.4 - 5.9 x 105 M-1". Should be 10 to the 5, not 105.

Answer:

We correct the typo ,,2.4 - 5.9 x 10-5 M-1’’.

7- In the docking section, it is very remarkable (and surprising) that a single methylene leads to an inversion in the binding mode between 1 and 2 (figure 5, a and b). Explain with more detail and add justifications.

Answer:

Thank you for this remark, this is indeed very interesting phenomena and we have added separate paragraph in the Molecular docking section explaining it. We have also added one figure and table in the Supporting Information file, to make the explanation more clear.

8- In the experimental section, please, give the yields with significative figures (42%, line 234, and all the rest)

Answer:

We give the yields of all compounds in experimental section.
